

# Ionospheric and thermospheric response to the 27-28 February 2014 geomagnetic storm

Khalifa Malki[1], Aziza Bounhir[1], Zouhair Benkhaldoun[1], Jonathan J. Makela[2], Nicole Vilmer[3], Daniel J. Fisher[2], Mohamed Kaab[1], Khaoula Elbouyahyaoui[1], Brian J. Harding[2], Amine Laghriyeb[1], Ahmed Daassou[1], and Mohamed Lazrek[1]

[1]Oukaimeden Observatory, High Energy Physics and Astrophysics Laboratory,FSSM, Cadi Ayyad University, Marrakesh BP : 2390, Morocco,
[2]Department of Electrical and Computer Engineering, University of Illinois at Urbana-Champaign, Urbana, Illinois, USA,
[3]LESIA,Observatoire de Paris, 5 Place Janssen, 92195-Meudon Cedex, France.

**Correspondence:** Khalifa Malki (malki.khalifa@gmail.com)

**Abstract.** The present work explores the ionospheric and thermospheric responses to the 27-28 February 2014 geomagnetic storm. For the first time, a geomagnetic storm is explored in north Africa using interferometer, all-sky imager and GPS data. This storm was caused by coronal mass ejection (CME) associated flares that occurred on 25 February 2014. A Fabry-Perot interferometer located at the Oukaimeden Observatory (31.206°N, 7.866°W, 22.84°N magnetic) in Morocco provides mea-

surements of the thermospheric neutral winds based on the observations of the 630 nm redline emission. A wide angle imaging system records images of the 630-nm emission. The effects of this geomagnetic storm on the thermosphere are evident from the clear departure of the neutral winds from their seasonal behavior. During the storm, the winds experience an intense and steep equatorward flow from 21 to 01 LT and a westward flow from 22 to 03 LT. The equatorial wind speed reaches a maximum of 120 m/s for the meridional component at 22 LT, when the zonal wind reverses to the westward direction. Shortly after 00

LT a maximum westward speed of 80 m/s was achieved for the zonal component of the wind. The features of the winds are typical of TAD (Traveling Atmospheric Disturbances) induced circulation; the first TAD coming from the northern hemisphere reaches the site at 21 LT and a second one coming from the southern hemisphere reaches the site at about 00 LT. We estimate the propagation speed of the northern TAD to be 550 m/s. We compared the winds to DWM07 (Disturbance Wind Model) prediction model and find that this model gives a good indication of the new circulation pattern caused by storm activity, but

deviates largely inside the TADs. The effects on the ionosphere were also evident through the change observed in the background electrodynamics from the reversal in drift direction in an observed equatorial plasma bubble. TEC measurements of a GPS station installed in Morocco, at Rabat (33.998°N; 6.853°W, geographic) revealed a positive storm.

## 1    Introduction

The sun is the energy provider of our planet through electromagnetic radiations, solar wind, and interplanetary magnetic field.

It is a highly variable star with sporadic events consisting of outbursts of huge amounts of energy. Solar flares are an impulsive release of a large amount of radiant energy that change in a very small time scale the physical properties of the ionosphere



which has a big impact on telecommunication and navigation. Coronal mass ejections (CMEs) are explosive outbursts of billion tons of plasma from the corona. CMEs carry powerful magnetic fields, shock waves and mostly protons and electrons. CMEs take from 1 to 4 days to travel to Earth, and interact with the solar wind and the interplanetary magnetic field (IMF) during their propagation, and can trigger geomagnetic storms when they interact with the Earth's magnetic field. Solar Particle Events

are a release of relativistic charged particles, mainly protons and electrons. A large coronal hole facing the earth can also cause a geomagnetic storm. Many changes occur in the magnetosphere when an intense surge of solar wind reaches the Earth. The geomagnetic field can become highly variable, and all the currents flowing through the magnetosphere and the ionosphere change rapidly. In response, the composition and dynamics of the ionosphere and the thermosphere are highly affected.

Space weather is related to the activities of the sun that can affect human beings on Earth. Our modern society depends on
technological tools for telecommunication, navigation, positioning, space exploration and energy provision. Solar events can cause satellite damage, radiation hazards to astronauts and airline passengers, telecommunication problems, outages of power and electronic systems and then endangering human life. As the importance of technology will inevitably increase in our daily lives there is a need to thoroughly understand the Sun/Earth system and space weather. The International Space Weather Initiative (ISWI), a program of international cooperation, was created to advance space weather science by a combination of
instrument deployment, analysis and interpretation of space weather data.

It is within the framework of the ISWI program that the RENOIR (Remote Equatorial Nighttime Observatory of Ionospheric Region) experiment was deployed in Morocco on November 2013 at The Oukaimeden Astronomical Observatory of the University Cady Ayyad ($31.206°N$, $7.866°W$, $22.84°N$ magnetic). The RENOIR experiment consists of a Fabry−Perot (FPI) interferometer and wide angle viewing camera. The FPI makes measurements of the thermospheric neutral wind veloc-
ities and neutral temperatures using observations of the 630.0 nm emission caused by the dissociative recombination of $O_2^+$. The wide−angle imaging system uses the same airglow emissions to provide measurements of ionospheric structures and irregularities. The main goal of this experiment is to characterize the mid-latitude ionosphere and thermosphere by establishing the climatology of both the neutral winds and the instabilities taking place in the ionosphere as well as the coupling between the ionosphere and thermosphere during quiet time conditions and during geomagnetic storms.

Thermospheric winds are driven primarily by pressure gradients due to absorption of solar radiation and collision between atmospheric constituents and precipitating auroral particles. During a geomagnetic storm, the energy input modify the global circulation of the thermosphere which profoundly influence the structure and composition of the ionosphere. Understanding the thermosphere/ionosphere coupling is the most challenging problem in space weather. It is within this context that we conduct thermospheric and ionospheric measurements over a mid-latitude site in north Africa by using data from an interferometer, a
GPS station and an all-sky imager. In the present paper, we analyse the ionspheric and thermospheric response to the 27-28 February 2014 geomagnetic storm. We illustrate the evident departure of the neutral winds from their normal behavior. A reversal in the background ionospheric electric field was evident through the dynamics of the plasma bubbles that occurred that night. The drift velocity of the EPB are compared to the zonal neutral winds in order to investigate the dynamo. TEC measurements precise the nature of the ionospheric storm. We have also conducted a comparison of the neutral winds with the
DWM model predictions. This is the first time that a case study of a geomagnetic storm has been achieved in north Africa by



using Fabry-Perot interferometer data of the thermosphere. Additionally, this is the first coincident study of a storm using a ground-based FPI and an all-sky imager in this sector.

## 2 The Interplanetary conditions

On 25 Feb 2014, an X-class flare associated with a coronal mass ejection and an energetic particle event took place. The CME associated shock reached ACE satellite on 27 Feb 16:30 UT with an increase in the density, temperature, velocity of the solar wind, the amplitude of the interplanetary magnetic field and an inversion of Bz magnetic field component. The resultant geomagnetic storm was characterized by the variations of the Kp, Dst, and AE geomagnetic indices shown in Figure 1. The Dst index (ring current index) is indicative of the total energy content of the particles responsible for the ring current. Geomagnetic storms with $|Dst|_{max}$ between 100 and 200 nT are classified as intense, with $|Dst|_{max} > 200$ nT as super-intense and other events with $|Dst|_{max}$ between 50 and 100 nT as moderate (Singh et al. (2017)). According to this classification, a moderate storm took place as observed during 27 Feb 2014 at $\sim$16:30:00 UT with $|Dst|_{max} \simeq$90 nT associated with the CME shock arrival. The AE index variations, based on high-latitude magnetograms (AE $\sim 400-500$ nT weak geomagnetic activity and AE $\sim 1000$ nT intense geomagnetic activity), also show a moderate storm activity on 27 Feb 2014 at 16:30 UT, with the AE index attaining values greater than 800 nT. The NOAA real-time observed $K_p$ index reached 5 during the synoptic period 16:00-18:00 UT on 27 February associated with the CME shock arrival. These intense variations in all geomagnetic indices are due to the arrival and interaction with Earth's magnetosphere of the coronal mass ejection (CME) from active region NOAA 11990. As the energy is deposited into the Earth's geospace system, effects are expected to propagate from high to low latitudes, modifying the Earth's thermopshere/ionosphere system. Below, we use measurements obtained from an observatory in Morocco to study these changes at middle latitudes.

## 3 Data and Instrumentation

Figure 2 shows the locations of the instruments used to collect data from the ionosphere and thermosphere. These instruments consist of a Fabry−Perot Interferometer (FPI) and a wide angle camera (Makela et al. (2011)) located at the Oukaimeden Observatory in Morocco (31.206°N, 7.866°W, 22.84°N magnetic, 2700 m of altitude). The FPI alternates between observations of the thermosphere at locations 250 km to the east, west, north and south of the observatory, and the viewing area of the camera covers a circle with $\sim$500 km radius. The city of Rabat (33.998°N; 6.854°W) where the GPS is located is included in that viewing area.

The FPI measures the Doppler shifts and Doppler broadenings of the 630.0 nm spectral emission and infers the thermospheric wind and temperatures for an altitude around 250 km. To be able to collect incident light from any direction, a system of two parallel mirrors called Sky Scanner is used which points in six directions during each cycle (laser, Zenith, North, East, South and West). The incident light passes through a narrowband interference filter (630.0 nm) and an etalon before beeing focused onto the CCD plane by a lens. An interference pattern is captured by the CCD and processed to produce estimates of the neutral



wind velocity and temperature Harding et al. (2014)). A frequency−stabilized HeNe laser is used to provide an estimate of the instrument's optical transfer function and to characterize the instrument's drift over the night.

The All-Sky Imaging System consists of a lens system, a five position filter wheel and a CCD device. One filter is used to isolate the 630-nm emission, allowing for visualizing plasma bubbles, medium−scale traveling ionospheric disturbances,

gravity waves and ionospheric irregularities in general (Makela and Miller (2008), Duly et al. (2013)).

Numerous studies have shown that the GPS is an effective tool for characterizing the ionosphere through measuring the Total Electron Content (TEC), which represents the total number of electrons integrated along the path from receiver to each GPS satellite (Ouattara et al. (2011); Sethi et al. (2001); Boutiouta and Belbachir (2006); Chauhan and Singh (2010)). Here, we present the Vertical Total Electron Content (VTEC) over Rabat (33.998°N; 6.853°W, geographic) obtained from the Interna-

tional GPS Geodynamics Service (IGS) network. The GPS data needed to determine the TEC are; 1)- data extracted from files stored in the RINEX (Receiver INdependent EXchange) format, 2)- data in IONEX format (IONosphere EXchange Format), 3)- Almanacs of the GPS satellite position and 4)- The satellite clock biases. The procedure for extracting the TEC from GPS data is well documented in the literature (Christian et al. (2013); Zoundi et al. (2012); Schaer (1999); Sardon et al. (1994); Klobuchar (1996)). In this analysis, we have extracted the TEC using a software developed by (Fleury et al. (2010); Fleury

et al. (2015))

## 4  Thermospheric and Ionospheric Response

### 4.1  Thermospheric wind Response

The day-night difference in solar heating and upward propagating atmospheric tides control the thermospheric wind circulation during quiet time conditions. During geomagnetic storms, joule heating is enhanced by strong high latitude electric currents

and by collisions of neutrals with convecting ions. These highly variable sources cause disturbances to thermospheric winds which will further affect the global ionospheric dynamo (Blanc and Richmond (1980)). In disturbed geomagnetic conditions, thermospheric neutral winds exhibit large deviations from their quiet time climatological behavior and can drive large changes in the ionospheric plasma density, composition, temperature, and electrodynamics (Rishbeth and Edwards (1989) ; Richmond and Matsushita (1975); Fuller-Rowell et al. (1994) ; Buonsanto (1999); Mendillo (2006); ?; Emmert et al. (2004)).

Before commenting on thermospheric winds behaviour over Oukaimeden observatory during the studied period (25 to 28 February 2014), we first begin by summarizing their basic climatology (Kaab et al. (2017)). In summer time, the meridional winds are equatorward for the entire night, reaching a maximum speed of 75 m/s. A poleward component is present in winter, in the early evening hours until 21 UT. The peak in equatorward flow shifts throughout the year from 23 UT during the spring equinox, to 02 UT during the autumn equinox. The zonal winds are eastward during the entire night with typical speeds around

50 m/s during the early evening hours, 75 to 100 m/s around 21 UT and almost zero before dawn in winter time. A westward reversal is present shortly before dawn in local summer.

Figure 3 shows the meridional and zonal winds measured with the Fabry−Perot interferometer (FPI) during 24, 25, 27 and 28 Feb 2014. The 26 Feb measurements are missing due to a power outage. The FPI data have been binned into half hour



bins. The error bars represent variability (mean$\pm\sigma$) within each 30 minute time bin. The average quiet time derived from measurements is in red color and the quiet time daily variability is illustrated as the shaded purple area. Positive wind values are respectively northward for the meridional winds and eastward for the zonal ones. Quiet time refers to a geomagnetic Kp index lower than four (Kp<4). We can clearly notice that the effect of the storm on 27 Feb 2014 is very noticeable on both

the meridional and zonal winds that has largely departed from their quiet time climatology. Figure 1 indicates that Kp is low until ~17:00 UT on 27 Feb, when Kp grows to 4 and eventually reaches 5. The meridional wind was equatorward and increase sharply at 21 UT to speed of about 140 m/s and reverse sharply to poleward direction around 01 UT. They remain poleward with low speed to 04 UT where they reverse again to equatorward direction. Instead of being eastward during the entire night, the zonal wind (on 27 Feb) starts eastward on the early evening with lower speeds than the quiet time climatology and reverse

westward at 22 UT achieving a maximum speed of 100 m/s around 00 UT and reverse back to eastward around 03 UT.

The steep and abrupt changes in the meridional winds is the signature of thermospheric traveling atmospheric disturbances flowing over the area covered by the Oukaimeden FPI measurements. The first TAD coming from the northern hemisphere was captured around 21 UT and the second TAD coming from the southern hemisphere was captured around 00 UT. At this point we find it interesting to represent the separate components of the meridional wind. Figure 4 shows the meridional wind estimates

made looking to the north and south at an elevation angle of 45°. These measurements are separated from one another by 500 km. In less than an hour the northern component of the wind gained almost 150 m/s speed towards the equator. The effect of the TAD coming from the north pole lasted for about four hours during which the northern component has higher speed than the southern one. The southern component reacted to the TAD with a delay of approximately 15 min. Our estimation then of the speed of the TAD over the studied area is about 550 m/s. (Xiong et al. (2015)) used the superposed epoch analysis to

CHAMP zonal wind observations from 2001 to 2005 and estimated the typical propagation speed of TADs to be 610 m/s. From Figure 4, the meridional winds reacted to the trans equatorial southern TAD at 00h20mn. Its effect lasted for 3.5 hours during which the southern location had a larger speed than the northern one. From 22 to 00 UT, the equatorward wind encountered some steep reactive effects, especially from 23 to 00 LT where the speed of the northern meridional wind lost about 100 m/s in 20min. Usually the quiet time northward surge of the wind related to the MTM phenomenon occured during that month

(February 2014) on average between 22h30 and 00h30 UT with an average amplitude of 20 m/s. The northward surge that occured between 23 and 00 UT on 27 February 2014 might be due to the enhancement of the MTM phenomenon by the storm circulation.

The thermospheric response, induced by high latitude heat input drives a global circulation of winds flowing from high to low latitudes (Buonsanto (1989); Buonsanto (1990)) creating large scale atmospheric waves (Richmond (1979)) that propagate

on a global scale. Due to Coriolis forces a subsequent change in the zonal circulation flowing westward, takes place. The storm induced atmospheric waves propagates to low latitudes and into the opposite hemisphere. The local impression of this global picture of the storm was captured during the 27 February event (see Figure 3). Indeed, the poleward surge occurred at 21 UT and drives the zonal wind to flow to the west at 22 UT. This result of westward and equatorward winds during the magnetic storm is consistent with previous results obtained with ground based Fabry−Perot interferometers (Makela et al.

(2014); Zhang et al. (2015)). (Xiong et al. (2015)) use the CHAMP zonal wind observations from 2001 to 2005 to investigate



the global features of the disturbance winds during magnetically disturbed periods. They report that the disturbance zonal wind is mainly westward, and increases with magnetic activity and latitude. Emmert et al. (2004) report westward and equatorward nighttime disturbed winds at midlatitude.

Emmert et al. (2008) present an empirical wind model (DWM07) representing average geospace-storm-induced perturba-
tions of upper thermospheric neutral winds. In constructing their model, they used data from the Wind Imaging Interferometer on board the Upper Atmosphere Research Satellite, the Wind and Temperature Spectrometer on board Dynamics Explorer 2, and seven ground-based Fabry-Perot interferometers. The wind derived from their empirical model is dependent on three parameters: magnetic latitude, magnetic local time, and the 3-h Kp geomagnetic activity index. To quantify the geomagnetically disturbed thermosphere, they established a quiet time reference that they subtracted from each measurement Emmert et al.
(2004). Emmert et al. (2008) is an improved version of the model presented in Emmert et al. (2004). To compare with DWM07 model predictions, we removed the average quiet time from the wind on Feb 27 (Figure 5) and ran the model in our location for that storm. The model predicts westward disturbed zonal winds with maximum speed of $\sim$ 60 m/s from 19 UT until 23 UT. The disturbed zonal wind during the 27 Feb geomagnetic storm (Kp=5) over Oukaimeden is westward during the whole night but with maximum speed of 140 m/s around 00 UT. The speed is then much higher than expected by DWM07 empirical
model. The meridional disturbed wind is clearly equator ward (Figure 5) until 00 UT with a particularly steep increase starting at 21 UT reaching a maximum speed of 90 m/s at 22.5 UT. An abrupt decreasing meridional speed occurs shortly after 00 UT reversing the disturbed winds to northward direction at 01 UT. However, the predicted meridional winds are extremely low with a reversal time of 23 UT instead of 01 UT for our measurements. The general feature of the wind agrees with the model predictions but we measure faster speeds and the effects of traveling atmospheric disturbances were clearly noticeable
from our measurements and absente from the model predictions. DWM07 model predictions give a good indication of the new circulation pattern caused by storm activity, however it deviates from the measurements inside the TADs. TADs when propagating from polar to equatorial regions causes an uplift in the F-peak height (HmF2) which in turn has a subsequent effect on the increase of the F-peak density (NmF2). Therefore, large-scale traveling ionospheric disturbance, consisting of sequential rises and falls in HmF2 (NmF2) happen. Models of storm response must then include dynamical wind effects, in addition to
changes in the circulation pattern, in order to reproduce realistic ionospheric responses.

### 4.2    Ionospheric electron density Response

The estimated TEC measured from a GPS station installed in Rabat (33.998°N, 6.854°W, geographic) is illustrated in Figure 6. This figure contains the TEC for the 27, 28 February 2014 and the Feb 2014 quiet days average (Kp<3). We can divide the quiet time shape mainly into three regions; region (1) from 07 to 13 UT where the TEC increases almost linearly from  15 to
45 TECU; region (2) from 13 to 00 UT where the TEC decreases almost linearly from 45 to 15 TECU and region (3) from 00 to 07 UT where the TEC is almost constant. The rate of the increase in region (1) is about 5 TECU/hour and the rate of the decrease in region (2) is about 2,7 TECU/hour. We can clearly notice the positive storm on 27 February with a maximum TEC of 65 TECU (around 13 UT), remaining high for some hours and achieving 50 TECU at 00 UT. On the early hours of 28, from 01 to 04 UT, the decrease is steep with an approximate rate of 8,3 TECU/hour. During the 28th, the TEC was still higher



than quiet time behavior. The observed difference during the early hours was about 40 TECU, and about 20 TECU during the day and diminishes to 10 TECU during the night. This result is consistent with previous observations of positive storms in the winter hemisphere and negative ones in the summer hemisphere (Fuller-Rowell et al. (1994), Fuller-Rowell et al. (1998)).

We observe oscillations of the TEC on the 27th between 14 UT and 00 UT and in the early hours on the 28th. As the meridional wind play an important role in the dynamics of the ionosphere, we present in Figure 7 the evolution of the TEC along with the meridional wind measured to the north and south. Only part of region (2) and region (3) are compared. In region (3), we observe a steep decrease of the TEC and an absence of correlation with the meridional winds. In this region, the TEC in quiet-time is normally almost constant at its lowest level. In storm time, between 00 and 06 UT, the TEC returns abruptly to its lowest level. Between 19 and 00 UT, we can notice that the TEC correlates in some ways with the meridional winds. In

fact, When the wind flow towards the equator, we observe a decrease of the TEC.

This observation contrasts with the quiet day-time behavior, in which a poleward wind drags the ionosphere down to regions of increased mean molecular mass, causing faster recombination and a reduced density (i.e., a negative correlation between poleward winds and plasma density). The reason a positive correlation is observed is likely due to transport caused by the TAD. At night, plasma production is near zero, yet increases in TEC are seen. This can only be due to transport. This observation

is consistent with previous studies discussing the negative correlation between hmF2 and NmF2 during the passage of TADs, at least in the initial phase (Bauske and Prölss (1997) and Lee et al. (2002)).Lee et al. (2004) used ionosondes, ROCSAT-1 satellite data and the Thermosphere Ionosphere Electrodynamics General Circulation Model (TIEGCM) to study the iono- spheric features of traveling atmospheric disturbances (TADs) during the 6-7 April 2000 magnetic storm. Their observations and corresponding TIEGCM simulations show a negative initial correlation between NmF2 and HmF2 caused by equatorward

wind surges at various locations from midlatitudes to the equator. A more detailed study of the processes involved with the TEC observations shown here would require co-located observations of hmF2 and NmF2.

### 4.3   Electrodynamic Response

Figure 8 shows a sequence of O I 630.0 nm airglow images illustrating the spatial characteristics and the time evolution of Equatorial Plasma Bubble (EPB) observed over Oukaimden from 23:02 LT to 03:58 LT during the night of 27-28 February

2014. The dark structures correspond to EPB signatures. While bubble motion is eastward in the early evening of 27 February, the plasma bubbles drift westward on 27 Feb 22 UT. This time of the reversal of the plasma drift corresponds to the time when the zonal wind starts reversing westward. This reversal of the drift direction of the plasma bubbles is a consequence of the reversal of the background electric field direction as driven by the neutral wind disturbance dynamo. This is consistent with both simulation and observational results (Blanc and Richmond (1980); Fejer and Emmert (2003)). However, these are the first

joint observations of EPBs during storm times from both a ground-based FPI and imager over north Africa.

According to (Blanc and Richmond (1980) during a magnetic storm both electric field and current at low latitudes vary in opposition to their normal quiet-day behavior in a sequence of electrodynamic phenomenon. During a geomagnetic storm a Hadley cell is created with equatorward and westward winds which in turn drive equatorward Pedersen currents resulting in



the generation of a poleward electric field, a westward E×B drift, and an eastward current which in turn create an anti-Sq type of current vortex.

An EPB drift velocity can be estimated using cross correlations between slices from the keograme. The slices are interpolated onto a uniformly spaced grid with a sub-pixel resolution of 1 km. The cross-correlation is taken between every $i^{th}$ image and

$i+5^{th}$ image and the lag of the correlation then produces a velocity. The five-image separation is chosen to limit the velocity uncertainty to ± 1 m/s. The drifts estimated throughout the night can then be compared to the zonal neutral winds to investigate the dynamo during storm time conditions. Figure 9 shows the estimated plasma drifts, measured zonal winds, and the quiet time monthly average zonal winds for this storm. Two major observations can be drawn from this figure; first, the neutral winds are indeed westward, indicating forcing from the geomagnetic storm, and second, the EPB drifts tend to closely match

the neutral winds, indicating that the dynamo is fully activated during this storm.

## 5    Summary and Conclusion

The effect of the 27 February 2014 geomagnetic storm on the thermosphere and the ionosphere is discussed in this paper. Thermospheric winds were inferred from Fabry-Perot interferometer measurements deployed at Oukaimeden observatory (31.206°N, 7.866°W, 22.84°N magnetic). The ionospheric response of the storm was also analyzed with the VTEC data

and a wide-angle imaging camera. Attention was paid to the conditions on the sun, interplanetary medium and geomagnetic indices that impacted the thermosphere/ionosphere coupling during that storm.

On February 25, an X-class flare accompanied with a coronal mass ejection occurred shortly after 00 UT. High energetic particles event was triggered at that time and lasted for more than three days. The geomagnetic storm occurred on 27 February at ∼16h 20 mn UT, according to interplanetary medium parameters of the shock waves and geomagnetic indices.

The zonal and meridional components of the winds were compared with their quiet time climatology to study the effect of the storm on the wind dynamics. Traveling atmospheric disturbances were evident in the meridional winds; the first one coming from the northern hemisphere and the second one coming from the southern hemisphere. From the time delay in the response of the northern and southern components of the meridional wind, we estimate the speed of the northern TAD to ∼550 m/s.

We compared the storm induced components of the winds to DWM07 model and concluded that while the model gives a

good approximation of the wind pattern, it largely departs inside the TAD. Traveling atmospheric disturbances trigger traveling ionospheric disturbances and to predict the effect of the storm on the ionosphere a model must include the dynamics of the winds and the time since the storm started.

The VTEC response of the storm was positive as expected during this time of the year. Evidence of TID were noticeable from the TEC pattern and correlates with the meridional components of the wind until 01 UT. We have observed a negative

correlation between HmF2 and NmF2 during the passage of the TAD.

The ionospheric response of the storm was also analyzed through the observation of equatorial plasma bubbles. We observe a reversal in the drift direction which indicate a reversal of the back-ground electric field direction due to the neutral wind disturbance dynamo. This is the first time that a case study of a geomagnetic storm has been achieved in north Africa by

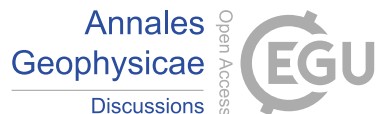

using Fabry-Perot interferometer data of the thermosphere. Additionally, this is the first coincident study of a storm using a ground-based FPI and an all-sky imager in this sector.





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



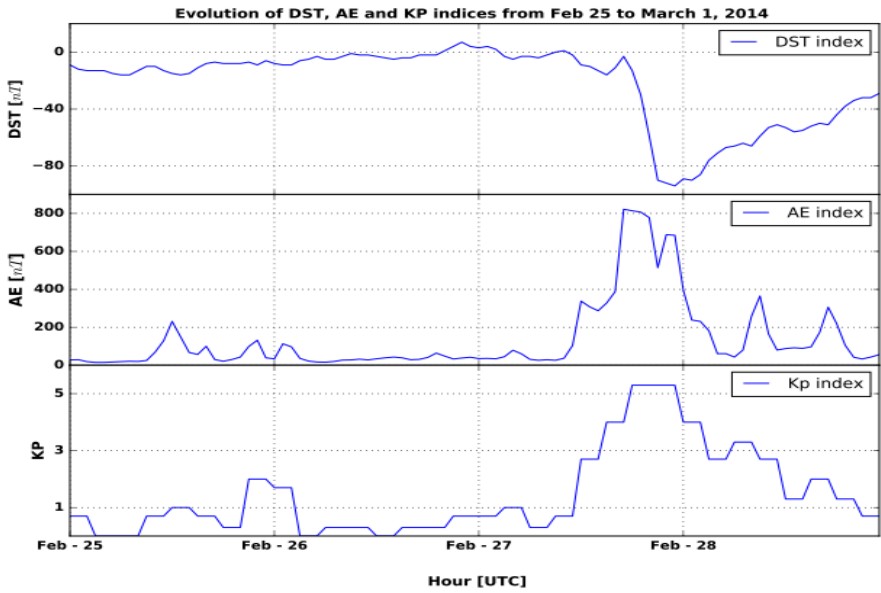

**Figure 1.** Time variations of the Kp, Dst and AE geomagnetic indices for the period 25−28 Feb 2014.

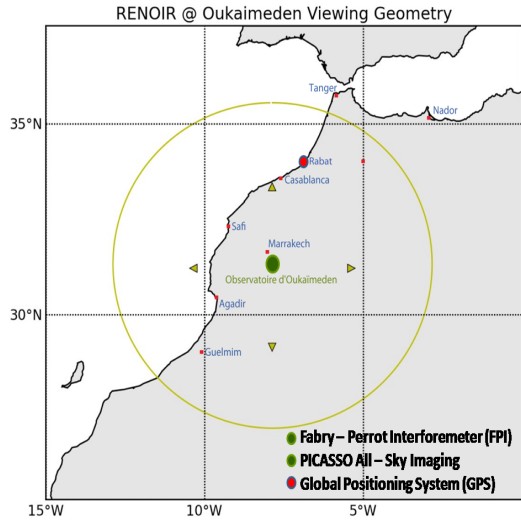

**Figure 2.** Viewing geometry (yellow circle) of the RENOIR experiment centered at Oukaimden Observatory (31.206°N, 7.866°W, 22.84°N magnetic). The city of Rabat (33.998°N; 353.1457°E) locating a GPS system is included in the viewing area. The yellow triangles indicate the viewing directions of the Fabry-Perot interferometer.





**Figure 3.** Thermospheric meridional and zonal winds measured by the the Fabry-Perot interferometer (FPI) over Oukaimeden Observatory (31.206°N, 7.866°W, 22.84°N magnetic). The FPI winds are estimated from the 630 nm airglow emission caused by dissociative recombination of $O_2^+$. The dates of measurements are the 24, 25, 27 and 28 Feb 2014. The red line refers to the monthly average of the nights with Kp strictly less than four. The blue line shows the meridional (zonal) measurements during the night and the magenta area denotes the daily variability during February 2014.





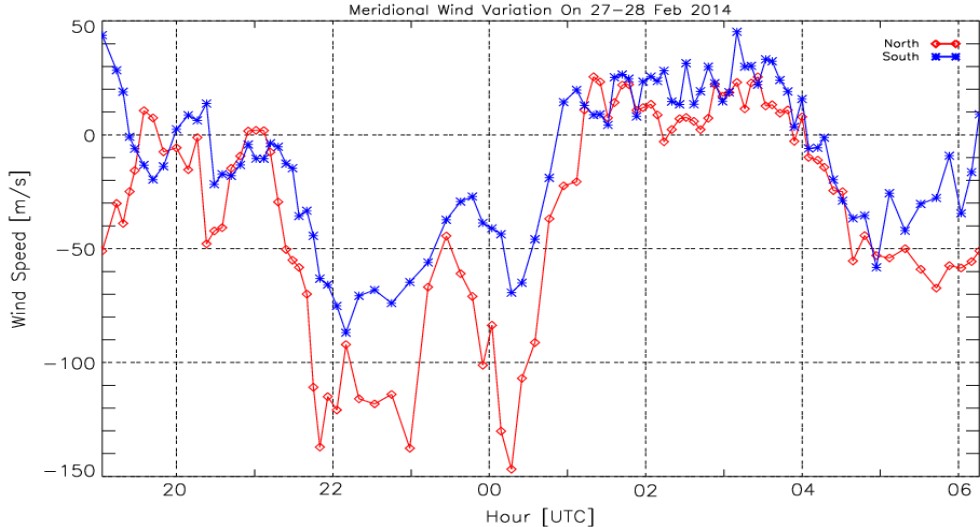

**Figure 4.** Thermospheric north and south look components of the meridional wind measured by the the Fabry-Perot interferometer (FPI) over Oukaimeden Observatory (31.206°N, 7.866°W, 22.84°N magnetic).





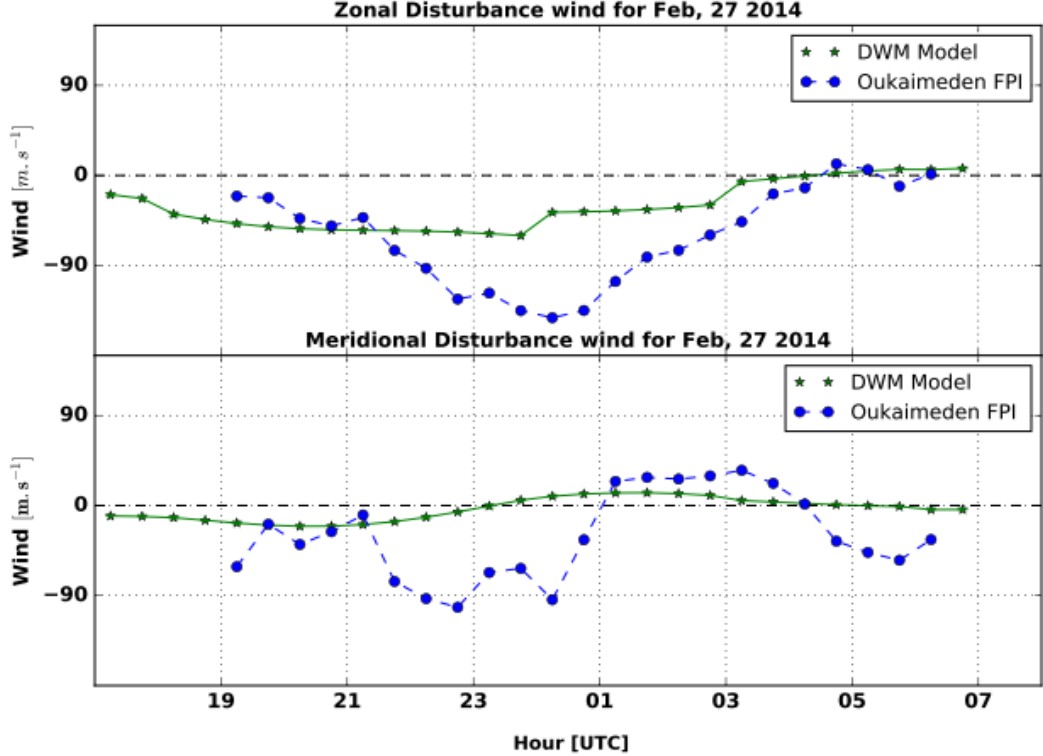

**Figure 5.** Comparison of zonal (Top) and meridional (Bottom) disturbance winds predicted by Disturbance Wind Model DWM07 (green stars) with disturbed thermospheric meridional and zonal winds obtained over Oukaimeden Observatory from which the monthly quiet time average ( with Kp<4 ) have been removed.

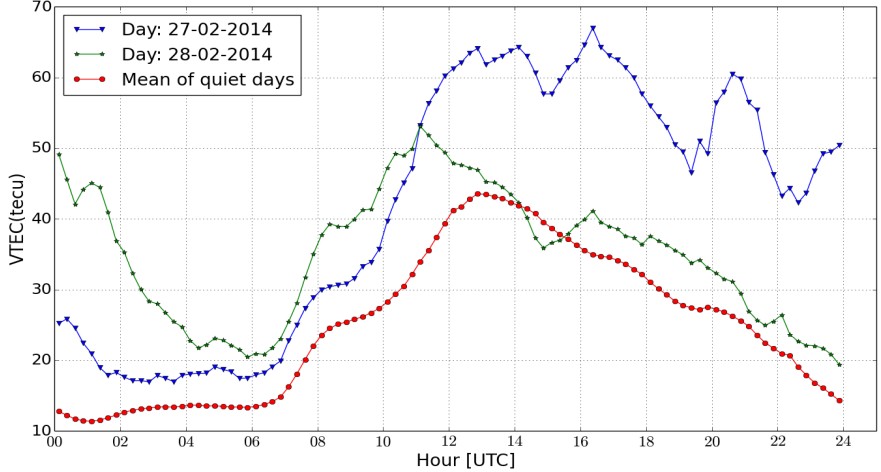

**Figure 6.** Total electron content TEC measured over Rabat (33.998°N; 6.854°W, geographic).



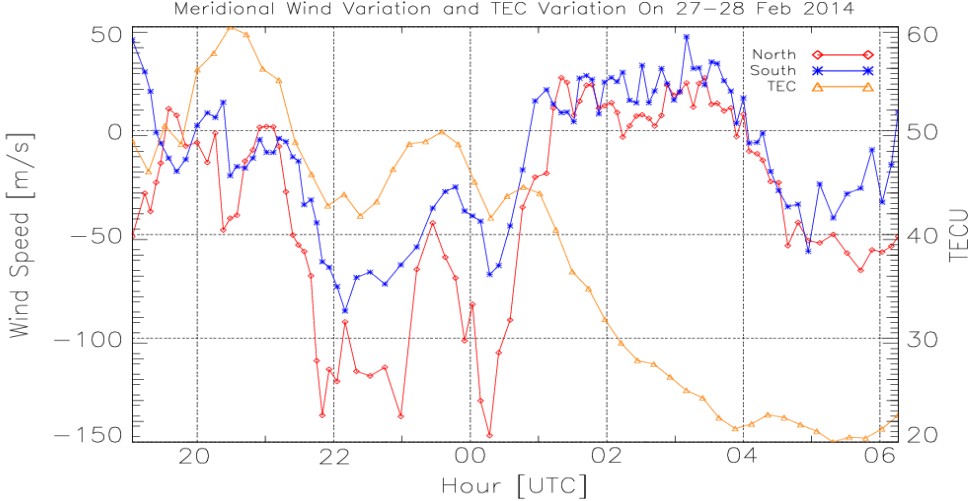

**Figure 7.** Total electron content TEC measured over Rabat (33.998°N; 6.854°W, geographic) and thermospheric meridional wind north and south look components, measured by the Fabry-Perot interferometer (FPI) over Oukaimeden Observatory (31.206°N, 7.866°W, 22.84°N magnetic).

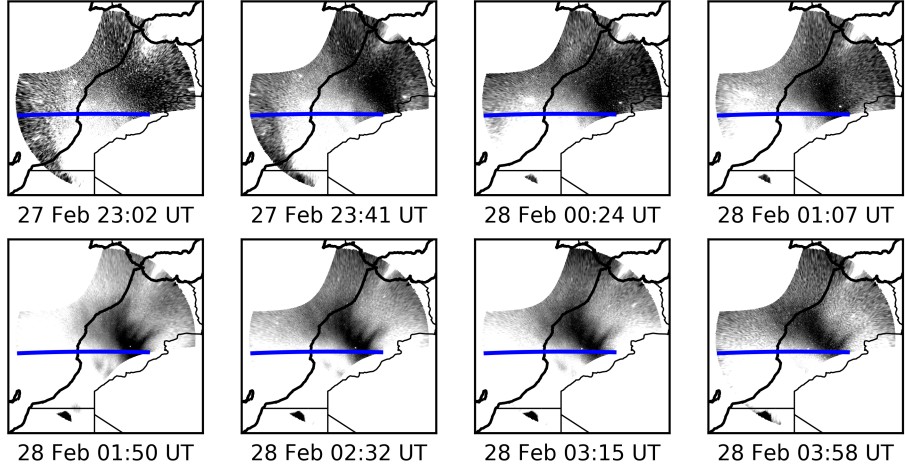

**Figure 8.** Sequence of O I 630.0 nm images showing spatial characteristics and time evolution of EPBs from 21:00 LT to 02:52 LT on 27-28 February 2014.




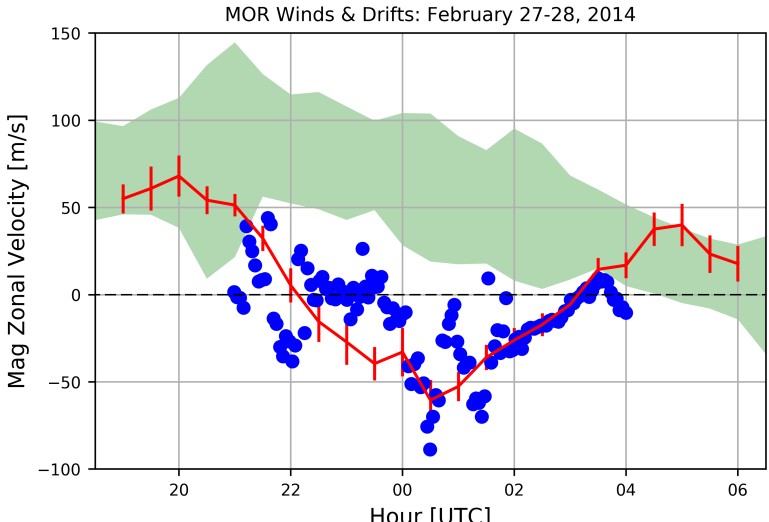

**Figure 9.** Estimated plasma drifts and neutral winds for the night of 27 February 2014. The plasma drift estimates from cross-correlations are marked as blue dots. The zonal neutral winds are plotted in red with error bars for the measurement uncertainties. The green shaded region displays typical range of quiet-time zonal neural winds for February 2014.