# Peer review of "Ionospheric and thermospheric response to the 27-28 February 2014 geomagnetic storm over north Africa"

_Annales Geophysicae, 2018_

## Referee Comment (RC1) · Anonymous Referee #1 · 16 Apr 2018

Dear Editor, Overall this is a very good article describes observations by a suite of the instruments during 27-28 February 2014 geomagnetic storm. It is also the first paper using such observations in Africa. I have only minor comments about the article.

L19 "The day-night difference in solar heating and upward propagating atmospheric tides control the thermospheric wind circulation during quiet time conditions. " Solar heating in the thermosphere is the main cause for the diurnal variation of the thermospheric winds. Tides can modulate the thermospheric wind (e.g. nonmigrating tides). But they are not the major source for wind circulation in the thermosphere. The sentence places the solar heating and tide in equal weight is not accurate.

L24 'Mendillo (2006); ?; Emmert et al. (2004))."' revise.

[Figure]

There should be some mention of the weather condition for the FPI observation.

Hope in the future, there will be some model comparisons.

---

## Author Comment (AC1) · 24 Apr 2018

Dear referee, Thank you very much for your encouraging remarks. Yes, you are right, the sentence "L19/page4: The day-night difference in solar heating and upward propagating atmospheric tides control the thermospheric wind circulation during quiet time conditions" can be misleading and understood as if the solar heating and tide are put in equal weight which is not accurate, as you said. Of course, the solar heating is the primary source that drives the thermospheric winds. Tides play secondary role compared to solar heating but important though as it modulates the wind circulation and deposit energy into the thermosphere. For example, nonmigrating tides shape the longitudinal dependence of the law-latitude ionosphere and an additional important effect is the upward propagating global wave features from the atmosphere below. Upward

propagating planetary waves, tides and gravity waves deposit in the thermosphere an important amount of energy as their amplitude grow with altitude and reach a point where they break down. It has been shown that thermospheric winds are heavily influenced by upward propagating tides and that they have a significant impact on the day-to-day variability of the winds. In regard to space weather tides are important. We therefore propose to change the sentence mentioned above by the following sentence: " L19/page4: The day-night difference in the solar heating is the primary source that control the thermospheric wind circulation during quiet time conditions. Another important secondary source is upward propagating tides."

Concerning the remark about the missing reference (L24/page4) here is the answer: Mendillo (2006); Meriwether (2008); Emmert et al.(2004)

Concerning the weather conditions, we have a cloud sensor. The sky has to be clear before measurements.

Concerning the comparison with models, we have compared the winds with the DWM model (see L23/page6).

---

## Referee Comment (RC2) · Anonymous Referee #2 · 2 May 2018

Title: Ionospheric and thermospheric response to the 27–28 February 2014 geomagnetic storm

This paper reports the response of F-region thermosphere and ionosphere over north Africa to the 27-28 Feb 2014 geomagnetic storm using observations based on an all sky imager, FPI, and GPS data. The all sky imager and FPI are collocated at the Oukaimeden Observatory (31.206N, 7.866W), while the GPS station is in Rabat (33.998N; 6.853W). This is the first case study of a geomagnetic storm from Africa using a co-located all sky imager and FPI. The strong departure of neutral winds from climatological winds reported in this study clearly demonstrates the control of geomagnetic activity on the low-latitude neutral winds. This study also reports the formation of an EPB that occurred at nighttime on 27 Feb and associated changes in ionospherethermosphere electrodynamics. Overall, this is a very straightforward study based on observations. The manuscript is well written and logically organized. On the whole this is a useful contribution to understand the local-scale ionosphere and thermosphere coupling dynamics under disturbed conditions.

I have few major and minor comments/suggestions. I would strongly recommend the authours to consider the following points and questions before publication:

The authour has shown neutral winds from 24 Feb to 1 March 2014 with primary focus on the storm day (27 Feb). The neutral winds on 27 Feb significantly differ from the non-storm day. The authour relates the variations in meridional winds on storm day to TADs and variations in zonal winds to EPB. Thus, an interplay of three different factors (geomagnetic storm, TADs, and EPB) affecting ion-neutral dynamics differently at the same time occurred on this day. They might not be independent of each other, but one can find days when geomagnetic activity was high and no TAD or EPB occurred or vice-versa. So, I would recommend the authour to include another case study when there was a geomagnetic storm, but no TAD or EPB appeared. That would eliminate the effect of change in electrodynamics associated with TADs or EPBs on the thermospheric winds above the observation region. This would isolate the pure effect of geomagnetic activity on neutral winds over the station from other factors.

I think the authour should include a location descriptor in the title (like - ..... geomagnetic storm over north Africa) as this is a study of localized events.

L32, page 2: Please include citation with "reversal in the background ionospheric electric field was evident through the dynamics of the plasma bubbles that occurred that night."

L33, page 2: Define EPB first and then start using this acronym.
L35, page 2: Replace DWM with DWM07 or stay consistent with the naming style.

L32, page 4: Figure 3 shows zonal and meridional winds measured with the FPI. As discussed in the section 3, this FPI measure LOS winds in four cardinal directions. So, you have 2 measurements in zonal and 2 in meridional direction. Did you average those LOS wind measurements in each direction to calculate zonal and meridional winds? If yes then please describe it either in section 3 or here.

Are they geographic or geomagnetic winds? Please state this too.

L33, page 4: The FPI measurements are binned into half hour bins. I am wondering why the FPI measurements are binned. The data could be shown at its original temporal resolution.

L1, page 5: The authour started using "quiet time" before defining it. It is defined in line 4, page 5.

L1, page 5: replace "The average quiet time derived from" to "The average quiet time wind components derived from".

L7, page 5: Instead of "sharply at 21 UT", it should be "sharply after 21 UT".

L9, page 5: replace "quiet time climatology and reverse" with "quiet time climatology and turn".

L11-12, page 5: Are these TADs associated with or caused by the geomagnetic storm? Please discuss.

L $\sim$ 15, page 5: Could these strong changes in zonal and meridional winds be related with the sharp changes in the background 630 nm airglow intensity as shown in Figure 8. The sharp intensity gradients in airglow in a small region in the FPI field of view may introduce errors in the recovered spectrum and those errors can propagate into the wind estimation procedure. Please discuss or cite.

L10-20, page 6: Please reorganize the sentences to make it more legible.

ANGEOD
L20, page 5: Please add some details regarding DWM07 that it is a climatological model. TADs do not occur periodically or are predictable. Therefore, their effects are washed out in climatology.

L28, page 6: Your definition of quiet-time has changed here - Kp

axis? Although this is not the focus of this paper, but it should be stated or discussed in the data and methodology section.

Also, what are MOR winds (in the title of Figure 9)?

---

## Short Comment (SC1) · 11 May 2018

Dear Referee,

Thank you very much for your very interesting remarks and valuable suggestions. In the attached file are the answear to your remarks. If there are any clarification that we could add, we will very glad to do so.

Best Regards, Aziza Bounhir

Please also note the supplement to this comment:
https://www.ann-geophys-discuss.net/angeo-2018-24/angeo-2018-24-SC1-supplement.pdf

---

## Author Comment (AC2) · 11 May 2018

Dear Referree,

Thank you very much for your remarks and suggestions.

…. So, I would recommend the authour to include another case study when there was a geomagnetic storm, but no TAD or EPB appeared. That would eliminate the effect of change in electrodynamics associated with TADs or EPBs on the thermospheric winds above the observation region. This would isolate the pure effect of geomagnetic activity on neutral winds over the station from other factors.

*1)- There are indeed many geomagnetic storms without the occurrence of EPB and TAD feature of the thermospheric wind. We have data of such nights but including them in this paper will modify the core of the paper which focuses on the 27-28 Feb geomagnetic storm.*

*EPB occur in quiet time [Makela et al., 2004] and disturbed nights [Huang et al., 2002]. TADs also occur in quiet and disturbed conditions [Fujiwara et al., 2006]. In quiet time, TADs are the origin of the Midnight Temperature Phenomena [Meriweather et al., 2011] which appears in our data as an enhancement of the thermospheric temperature and a reversal of the meridional winds. This is localized event in time. It is as if this localized event superposes to the quiet time feature of the winds and temperature. TAD resulting from geomagnetic storms [Ritter et al., 2010] have totally different features from those occurring in quiet time. The TAD that occurred on the 27 Feb 2014 is caused by the storm. There might be a minor contribution of what we could call "a quit time TAD" superposed to the "the storm TAD".*

*The fact that on that night EPB develop was helpful to see the electrodynamics of the ionospheric medium.*

*Thank you for this relevant remark. There are indeed important questions to address in regard to our data; 1) do EPB develop equally in quiet and disturbed nights? 2) what is the rate of development of TAD during geomagnetic storms. It is important to make a statistical study to answer those questions.*

I think the authour should include a location descriptor in the title (like - …………….. geomagnetic storm over north Africa) as this is a study of localized events.

*2)- We will change the tittle to* **"Ionospheric and thermospheric response to the 27–28 February 2014 geomagnetic storm of the western part of north Africa"**

Line 9 page 1: I think replace "……… 22 LT, when the zonal….." with "……… 22 LT, after the zonal….."

*3)- We correct in the text.*

L32, page 2: Please include citation with "reversal in the background ionospheric electric field was evident through the dynamics of the plasma bubbles that occurred that night."

*4)- We added those citations at the end of the phrase L33, P2. [B.G. Fejer and al., 2016, Blanc and Richmond, 1980]*

L33, page 2: Define EPB first and then start using this acronym.

*5)- We define EPB; Equatorial Plasma Bubbles in the text.*

L35, page 2: Replace DWM with DWM07 or stay consistent with the naming style.

*6)- We change in the whole paper DWM to DWM07.*

L32, page 4: Figure 3 shows zonal and meridional winds measured with the FPI. As discussed in the section 3, this FPI measure LOS winds in four cardinal directions. So, you have 2 measurements in zonal and 2 in meridional direction. Did you average those LOS wind measurements in each direction to calculate zonal and meridional winds? If yes then please describe it either in section 3 or here.

*7)- Yes, this is right we averaged the meridional LOS and the zonal LOS.*

*We added in L33, P 4;"…power outage. The FPI Measures Line of Sight (LOS) winds in four cardinal directions; East, West, North and South. The East and West LOS winds are zonal winds and the North and South LOS winds are meridional winds.*

Are they geographic or geomagnetic winds? Please state this too.

*The winds are geographic, we included this precision in the text. L32, P4 we changed the phrase to "Fig 3 shows the geographic meridional and zonal winds …."*

L33, page 4: The FPI measurements are binned into half hour bins. I am wondering why the FPI measurements are binned. The data could be shown at its original temporal resolution.

*8)- Yes, you are right. The idea was to have a component for the meridional wind and one for the zonal one. The FPI makes a sequence of measurements (laser, Zenith, North, East, South and West), one after another. In order to have a meridional and a zonal component of the winds we have to average. The meridional look directions are separated by 500 km and so are the zonal ones. In a previous paper dedicated to the climatology of the winds over our region we followed this protocol [Kaab et al., 2017].*

 *This was also useful for the comparison with the DWM07 model. To compute the disturbance winds in Figure 5, we needed to compute the climatology and disturbed winds on the same time basis. Therefore, some type of interpolation or binning was necessary for that analysis. In Figure 4 and 7 we show the original temporal resolution and show separately the North and South look directions.*

L1, page 5: The authour started using "quiet time" before defining it. It is defined in line 4, page 5.

*9)- We will define the quiet time earlier; in line 1 page 5 we will add "…..30 minute time bin. The quiet time refers to data with 3 hours Kp < 4. The average quiet time wind components derived from …."*

L1, page 5: replace "The average quiet time derived from" to "The average quiet time wind components derived from".

*10)- We correct in the text.*

L7, page 5: Instead of "sharply at 21 UT", it should be "sharply after 21 UT". L9, page 5: replace "quiet time climatology and reverse" with "quiet time climatology and turn".

*11)- We correct in the text.*

L11-12, page 5: Are these TADs associated with or caused by the geomagnetic storm? Please discuss.

*12)- These TADs are caused by the storm given their features. (Please see answer 1)*

 *TADs are caused by geomagnetic storms [Bruinsma and Forbes, 2006; Jiuhou, et al., 2008] and sub-storms [Ritter et al., 2010]. During storms and sub-storms, the thermosphere density*

*at high latitude is enhanced and the disturbance travels to lower latitudes with typical speed of 650m/s and reaches the equator 3-4 hours later on average. The wind features related to those TADs are absent in our quiet time data. TADs occur also in quiet time and are responsible for the MTM phenomena which exhibit totally different wind and temperature patterns.*

L~15, page 5: Could these strong changes in zonal and meridional winds be related with the sharp changes in the background 630 nm airglow intensity as shown in Figure 8. The sharp intensity gradients in airglow in a small region in the FPI field of view may introduce errors in the recovered spectrum and those errors can propagate into the wind estimation procedure. Please discuss or cite.

*13)- Figure 8 shows that there are large-scale gradients in the airglow brightness over Oukaimeden. These do not affect the wind measurements shown here because the FPI has a field of view subtending a ~10 km diameter at 250 km altitude. Significant airglow gradients are not expected in this small area, especially gradients that persist over multiple minutes of exposure time. Large-scale gradients outside the field of view are a potential concern [Harding et al., 2017], but the situation here does not meet the criteria for a large atmospheric scattering effect. Namely, there is not a bright region with a large line-of-sight wind (the dark region in the north here has the largest line-of-sight wind), and the vertical wind estimate does not appear to be affected by more than 30 m/s.*
*The contrast in figure 8 is not the real contrast between the background and the EPBs. We will add a video extracted from the images collected from the camera. We changed this figure by putting images of the EPBs that illustrate the reversal of the drift velocity of the EPBs. In this figure we will not change the contrast and you will see that the difference between the background and the EPBs is very faint.*

L10-20, page 6: Please reorganize the sentences to make it more legible.

*14)- We will reorganize the sentences; from L8-20 page6;" To quantify the geomagnetically disturbed thermosphere, they established a quiet time reference that they subtracted from each measurement [Emmert et al. (2008)]. We removed the average quiet-time wind from the wind measured on Feb 27 and compared this with the model prediction by DWM07 for this storm. Results are shown in Figure 5. The model predicts westward disturbed zonal winds with maximum speed of 60 m/s from 19 UT until 23 UT. The measured disturbance wind is westward during the whole night but with a maximum speed of 140 m/s around 00 UT. The speed is then much higher than expected by DWM07 empirical model. The meridional disturbed wind is clearly equator ward (Figure 5) until 00 UT with a particularly steep increase starting at 21 UT reaching a maximum speed of 90 m/s at 22.5 UT. An abrupt decreasing meridional speed occurs shortly after 00 UT reversing the disturbed winds to northward direction at 01 UT. However, the predicted meridional winds are extremely low with a reversal time of 23 UT instead of 01 UT for our measurements.*
*The general feature of the wind agrees with the model predictions but we measure faster speeds and the effects of traveling atmospheric disturbances were clearly noticeable from our measurements and absent from the model predictions".*

L20, page 5: Please add some details regarding DWM07 that it is a climatological model. TADs do not occur periodically or are predictable. Therefore, their effects are washed out in climatology.

*15)- We modified the text starting from L18 page 6; "The general feature of the wind agrees with the model predictions but we measure faster speeds, and the effects of TADs were clearly noticeable from our measurements and absent from the model predictions. The DWM07 model predictions provide a good indication of the new circulation pattern caused by storm activity, but it does not include the effects of TADs. This is expected because DWM07 is a climatological model. It does not include dynamics, so it can not model TADs."*

L28, page 6: Your definition of quiet-time has changed here - Kp<3. Earlier it was Kp<4.

*16)- We will stay consistent and keep the 3 hours Kp < 4 as the definition of quiet time in the TEC data and the FPI data.*

L7-9, page 7: Please remove redundancy.

*17)- We removed the redundancy and replace the phrase (L 7-9 page 8) by; "… absence of correlation with the meridional winds. In this region, on a normal day, the TEC is small and not changing during this time. On the storm day during this time, the TEC starts high but returns quickly to its normal values. In region 2, between 19 and 00 UT, we can notice that the TEC correlates in some ways with the meridional winds. In fact, when the meridional winds are more equatorward, we observe a decrease of the TEC."*

L10, page 7: I think you are trying to say here that "when meridional winds are more equatorward, ……."

*18)- Yes. We correct in the text.*

L12, page 7: "reduced density" - I think you mean "reduced plasma density".

*19)- Yes. We precise it the text.*

L25-26, page 7: I could not extract this information from Figure 8. For example, 27 Feb at 22 UT - this time period is not shown in Figure 8. An extended temporal sequence of EPB motion in Figure 8 is desirable to assist what you have described here.

*20)- We changed this figure by another one that illustrates the reversal of the drift velocity of the EPBs (without changing the contrast).*

L3, page 8: slices of keogram in zonal or meridional direction?

*21)- The keogram is made using geomagnetic zonal cuts. We will change the paragraph and we will not mention the keogram since we are not displaying it.*

L3-10, page 8: I would suggest to elaborate this technique more in this paragraph.

*22)- We changed the paragraph; L 3-10 by; "Figure 9 shows the EPB zonal drift velocity which is estimated from the images in Figure 8 by performing a cross-correlation analysis. We assume here that the EPB drift velocity is a good proxy for the background plasma drift velocity. Slices of the image are taken along a line of constant magnetic latitude (the blue line in Figure 8), and a cross-correlation is taken between each image and an image 15 minutes later. The time lag corresponding to the maximum correlation is used to determine a velocity, under the assumption that structures in the image do not change significantly over 15 minutes. The zonal drifts estimated throughout the night can then be compared to the zonal neutral winds to investigate the dynamo during storm time conditions. Figure 9 also shows the average measured zonal winds and quiet time monthly average zonal winds for this storm. Due to the small magnetic declination at Oukaimeden (~2 degrees), the magnetic and geographic winds are nearly identical, allowing this comparison. Two major observations can be drawn from this figure; first, the neutral winds are indeed westward, indicating forcing from the geomagnetic storm, and second, the EPB drifts tend to closely match the neutral winds, indicating that the dynamo is fully activated during this storm".*

L5-6, page 8: It would be nice to know how this uncertainty of +-1 m/s was calculated.

*23)- Given an assumed projection altitude of the redline emission of 250 km, the width of sky measured by one pixel of the CCD for the All-Sky Imager is roughly 1000 m. We chose to correlate with images separated by ~15 minutes, meaning the smallest observable drift speed estimated from this technique is +/- 1 m/s (this also assumes that the features in the sky do not vary significantly over time and we can perfectly track it moving over exactly one pixel.) Likewise, the maximum limit for observable drift speed from this technique is +/- 264 m/s (assumes we correlate using the center 66% of the pixels).*

*We have removed the discussion on the uncertainty from the text.*

L17-19, page 8: This information is redundant.

*24)- Yes. We eliminate L17, L18 and L19 in page 8.*

L29, page 8: until 01 UT of 29 Feb.

*25)- Yes. We correct in the text.*

Figure 3: Figure show zonal and meridional winds, but are they geographic or geomagnetic? Please describe. Also, their titles "Meridional wind vs Quiet time" and "Zonal wind vs Quiet time" are not appropriate. Please change them.

*26)- Figure 3 shows geographic meridional and zonal winds. We will precise in the figure legend that these winds are geographic. We have removed the title of figure 3.*

Figure 4: I think Figure 4 is redundant as Figure 7 includes all the information displayed in Figure 4. So, you can remove Figure 4 (if you want).

*27)- Yes, we will keep both.*

Figure 9: I assume the winds shown earlier are geographic. How did you calculate magnetic zonal winds from the LOS winds? Is the instrument aligned with magnetic axis? Although this is not the focus of this paper, but it should be stated or discussed in the data and methodology section.

*28)- Winds are geographic zonal winds because the declination is only two degrees over Morocco, so we assume that they are equivalent to the geomagnetic winds. This simplification was made because to properly calculate a geomagnetically zonal wind, we would need to rotate the zonal and meridional winds into the magnetic zonal direction. This then requires a meridional wind measurement for each zonal wind measurement; thus our zonal wind estimates would be smoothed over/ blurred because more assumptions must be made to combine N/S E/W measurements together (and this would increase uncertainties and decrease our time resolution of the winds).*

Also, what are MOR winds (in the title of Figure 9)?

*MOR is the name of Oukaimeden station in the network of FPIs. We will remove this name from this figure.*

**Bibliography;**

Blanc, M. and Richmond, A.: The ionospheric disturbance dynamo, Journal of Geophysical Research: Space Physics, 85, 1669–1686, 1980.

B.G. Fejer · M. Blanc · A.D. Richmond, Post-Storm Middle and Low-Latitude Ionospheric Electric Fields Effects, Space Sci Rev DOI 10.1007/s11214-016-0320-x.

Hitoshi Fujiwara and Yasunobu Miyoshi, Characteristics of the large-scale traveling atmospheric disturbances during geomagnetically quiet and disturbed periods simulated by a whole atmosphere general circulation mode, GEOPHYSICAL RESEARCH LETTERS, VOL. 33, L20108, doi:10.1029/2006GL027103, 2006.

Harding, B. J. , J. J. Makela, J. Qin, D. J. Fisher, C. R. Martinis, J. Noto, and C. M. Wrasse (2017), Atmospheric scattering effects on ground-based measurements of thermospheric vertical wind, horizontal wind, and temperature, *Journal of Geophysical Research: Space Physics, 122*(7), 7654 - 7669,

J. J. Makela, B. M. Ledvina, M. C. Kelley, and P. M. Kintner, Analysis of the seasonal variations of equatorial plasma bubble occurrence observed from Haleakala, Hawaii, Annales Geophysicae (2004) 22: 3109–3121

J. W. Meriwether, J. J. Makela, Y. Huang, D. J. Fisher, R. A. Buriti, A. F. Medeiros, and H. Takahashi, Climatology of the nighttime equatorial thermospheric winds and temperatures over Brazil near solar minimum, JOURNAL OF GEOPHYSICAL RESEARCH, VOL. 116, A04322, doi:10.1029/2011JA016477, 2011.

P. Ritter, H. L¨uhr, and E. Doornbos, Substorm-related thermospheric density and wind disturbances derived from CHAMP observation, Ann. Geophys., 28, 1207–1220, 2010.

C. Y. Huang, W. J. Burke, J. S. Machuzak, L. C. Gentile, and P. J. Sultan, Equatorial plasma bubbles observed by DMSP satellites during a full solar cycle: Toward a global climatology, JOURNAL OF GEOPHYSICAL RESEARCH, VOL. 107, NO. A12, 1434, doi:10.1029/2002JA009452, 2002

Kaab, M., Benkhaldoun, Z., Fisher, D. J., Harding, B., Bounhir, A., Makela, J. J., Laghriyeb, A., Malki, K., Daassou, A., and Lazrek, M.: Climatology of thermospheric neutral winds over Oukaïmeden Observatory in Morocco, Annales Geophysicae, 35, 161–170, 2017.

Sean L. Bruinsma and Jeffrey M. Forbes, Global observation of traveling atmospheric disturbances (TADs) in the thermosphere, GEOPHYSICAL RESEARCH LETTERS, VOL. 34, L14103, doi:10.1029/2007GL030243, 2007.

*Jiuhou Lei, Alan G. Burns, Takuya Tsugawa, Wenbin Wang, Observations and simulations of quasiperiodic ionospheric oscillations and large-scale traveling ionospheric disturbances during the December 2006 geomagnetic storm, JOURNAL OF GEOPHYSICAL RESEARCH, VOL. 113, A06310, doi:10.1029/2008JA013090, 2008.*

---

## Author Response (AR1)

Dear Editor,

We are very pleased to answer your request.

We have changed the paper according to suggestions from the reviewers. We have also included some additional changes that seem more appropriate especially in the introduction and section 2. The parts that have been changed are in blue color.

Following your recommendations, you will find answers to the first and second referee.

**Answers to referee 1;**

1)- L19 "The day-night difference in solar heating and upward propagating atmospheric tides control the thermospheric wind circulation during quiet time conditions. "Solar heating in the thermosphere is the main cause for the diurnal variation of the thermospheric winds. Tides can modulate the thermospheric wind (e.g. nonmigrating tides). But they are not the major source for wind circulation in the thermosphere. The sentence places the solar heating and tide in equal weight is not accurate.

Yes, you are right; the sentence "The day-night difference in solar heating and upward propagating atmospheric tides control the thermospheric wind circulation during quiet time conditions" can be misleading and understood as if the solar heating and tide are put in equal weight which is not accurate, as you said. Of course, the solar heating is the primary source that drive the thermospheric winds. Tides play secondary role compared to solar heating but important though as it modulates the wind circulation and deposit energy into the thermosphere. For example, nonmigrating tides shape the longitudinal dependence of the law-latitude ionosphere and an additional important effect is the upward propagating global wave features from the atmosphere below. Upward propagating planetary waves, tides and gravity waves deposit in the thermosphere an important amount of energy as their amplitude grow with altitude and reach a point where they break down. It has been shown that thermospheric winds are heavily influenced by upward propagating tides and that they have a significant impact on the day-to-day variability of the winds. In regard to space weather tides are important.

We therefore propose to change the sentence mentioned above (L19 page 4) by the following sentence: "The day-night difference in the solar heating is the primary source that control the thermospheric wind circulation during quiet time conditions. Another important secondary source are upward propagating tides."

2)- L24 'Mendillo (2006); ?; Emmert et al. (2004). *We have corrected this reference.*

3)- There should be some mention of the weather condition for the FPI observation. Hope in the future, there will be some model comparisons.

Concerning the weather conditions, we have a cloud sensor. The sky has to be clear before measurements. Concerning the comparison with models, we have compared the winds with the DWM07 model.

**Answers to referee 2;**

.... So, I would recommend the authour to include another case study when there was a geomagnetic storm, but no TAD or EPB appeared. That would eliminate the effect of change in electrodynamics associated with TADs or EPBs on the thermospheric winds above the observation region. This would isolate the pure effect of geomagnetic activity on neutral winds over the station from other factors.

1)- There are indeed many geomagnetic storms without the occurrence of EPB and TAD feature of the thermospheric wind. We have data of such nights but including them in this paper will modify the core of the paper which focuses on the 27-28 Feb geomagnetic storm.

EPB occur in quiet time [Makela et al., 2004] and disturbed nights [Huang et al., 2002]. TADs also occur in quiet and disturbed conditions [Fujiwara et al., 2006]. In quiet time, TADs are the origin of the Midnight Temperature Phenomena [Meriweather et al., 2011] which appears in our data as an enhancement of the thermospheric temperature and a reversal of the meridional winds. This is localized event in time. It is as if this localized event superposes to the quiet time feature of the winds and temperature. TAD resulting from geomagnetic storms [Ritter et al., 2010] have totally different features from those occurring in quiet time. The TAD that occurred on the 27 Feb 2014 is caused by the storm. There might be a minor contribution of what we could call "a quit time TAD" superposed to the "the storm TAD".

The fact that on that night EPB develop was helpful to see the electrodynamics of the ionospheric medium. Thank you for this relevant remark. There are indeed important questions to address in regard to our data; 1) do EPB develop equally in quiet and disturbed nights? 2) what is the rate of development of TAD during geomagnetic storms. It is important to make a statistical study to answer those questions.

I think the authour should include a location descriptor in the title (like - ..... geomagnetic storm over north Africa) as this is a study of localized events.

2)- We will change the tittle to "Ionospheric and thermospheric response to the 27–28 February 2014 geomagnetic storm over north Africa"

We proposed earlier the referee 2 this tittle;

*"lonospheric and thermospheric response to the 27–28 February 2014 geomagnetic storm of the western part of north Africa"*

Line 9 page 1: I think replace "....... 22 LT, when the zonal....." with "...... 22 LT, after the zonal....." 3)- We correct in the text.

L32, page 2: Please include citation with "reversal in the background ionospheric electric field was evident through the dynamics of the plasma bubbles that occurred that night."

4)- We added those citations at the end of the phrase L33, P2. [B.G. Fejer and al., 2016, Blanc and Richmond, 1980]

L33, page 2: Define EPB first and then start using this acronym.

5)- We define EPB; Equatorial Plasma Bubbles in the text. C2

L35, page 2: Replace DWM with DWM07 or stay consistent with the naming style.

*6)- We change in the whole paper DWM to DWM07.*

L32, page 4: Figure 3 shows zonal and meridional winds measured with the FPI. As discussed in the section 3, this FPI measure LOS winds in four cardinal directions. So, you have 2 measurements in zonal and 2 in meridional direction. Did you average those LOS wind measurements in each direction to calculate zonal and meridional winds? If yes then please describe it either in section 3 or here.

7)- Yes, this is right we averaged the meridional LOS and the zonal LOS.

We added in L33, P 4;"...power outage. The FPI Measures Line of Sight (LOS) winds in four cardinal directions; East, West, North and South. The East and West LOS winds are zonal winds and the North and South LOS winds are meridional winds.

Are they geographic or geomagnetic winds? Please state this too.

The winds are geographic, we included this precision in the text. L32, P4 we changed the phrase to "Fig 3 shows the geographic meridional and zonal winds ...."

L33, page 4: The FPI measurements are binned into half hour bins. I am wondering why the FPI measurements are binned. The data could be shown at its original temporal resolution.

8)- Yes, you are right. The idea was to have a component for the meridional wind and one for the zonal one. The FPI makes a sequence of measurements (laser, Zenith, North, East, South and West), one after another. In order to have a meridional and a zonal component of the winds we have to average. The meridional look directions are separated by 500 km and so are the zonal ones. In a previous paper dedicated to the climatology of the winds over our region we followed this protocol [Kaab et al., 2017].

This was also useful for the comparison with the DWM07 model. To compute the disturbance winds in Figure 5, we needed to compute the climatology and disturbed winds on the same time basis. Therefore, some type of interpolation or binning was necessary for that analysis. In Figure 4 and 7 we show the original temporal resolution and show separately the North and South look directions.

L1, page 5: The authour started using "quiet time" before defining it. It is defined in line 4, page 5.

*9)- We will define the quiet time earlier; in line 1 page 5 we will add "……30 minute time bin. The quiet time refers to data with 3 hours Kp

C. Y. Huang, W. J. Burke, J. S. Machuzak, L. C. Gentile, and P. J. Sultan, Equatorial plasma bubbles observed by DMSP satellites during a full solar cycle: Toward a global climatology, JOURNAL OF GEOPHYSICAL RESEARCH, VOL. 107, NO. A12, 1434, doi:10.1029/2002JA009452, 2002 Kaab, M., Benkhaldoun, Z., Fisher, D. J., Harding, B., Bounhir, A., Makela, J. J., Laghriyeb, A., Malki, K., Daassou, A., and Lazrek, M.: Climatology of thermospheric neutral winds over Oukaïmeden Observatory in Morocco, Annales Geophysicae, 35, 161–170, 2017.

Sean L. Bruinsma and Jeffrey M. Forbes, Global observation of traveling atmospheric disturbances (TADs) in the thermosphere, GEOPHYSICAL RESEARCH LETTERS, VOL. 34, L14103, doi:10.1029/2007GL030243, 2007.

Jiuhou Lei, Alan G. Burns, Takuya Tsugawa, Wenbin Wang, Observations and simulations of quasiperiodic ionospheric oscillations and large-scale traveling ionospheric disturbances during the December 2006 geomagnetic storm, JOURNAL OF GEOPHYSICAL RESEARCH, VOL. 113, A06310, doi:10.1029/2008JA013090, 2008.

---

## Author Response (AR2)

Dear Editor,

We are very pleased to answer your request related to video extracted from the images collected from the camera.

We have uploaded the file previously missed: picasso04_mor_20140227_6300movie.avi.zip

Sorry for this inattention.

On behalf off all co-authors

Malki Khalifa